# Impact of Mechanical Load on the Expression Profile of Synovial Fibroblasts from Patients with and without Osteoarthritis

**DOI:** 10.3390/ijms20030585

**Published:** 2019-01-30

**Authors:** Agnes Schröder, Ute Nazet, Dominique Muschter, Susanne Grässel, Peter Proff, Christian Kirschneck

**Affiliations:** 1Department of Orthodontics, University Hospital Regensburg, 93053 Regensburg, Germany; ute.nazet@ukr.de (U.N.); peter.proff@ukr.de (P.P.); christian.kirschneck@ukr.de (C.K.); 2Department of Orthopaedic Surgery, Experimental Orthopaedics, Centre for Medical Biotechnology, University of Regensburg, 93053 Regensburg, Germany; dominique.muschter@ukr.de (D.M.); susanne.graessel@ukr.de (S.G.)

**Keywords:** osteoarthritis, synovial fibroblasts, mechanical loading, synovium, inflammation, synovitis, compressive force

## Abstract

Osteoarthritis (OA) affects the integrity of the entire joint including the synovium. The most abundant cells in the synovium are fibroblasts (SF). Excessive mechanical loading might contribute to OA pathogenesis. Here, we investigate the effects of mechanical loading on SF derived from non-OA (N-SF) and OA patients (OA-SF). We treated N-SF and OA-SF with or without mechanical loading for 48h after 24h of preincubation. Then we assessed gene and protein expression of proinflammatory factors (TNFα, COX-2, PG-E2, IL-6), extracellular matrix (ECM) components (COL1, FN1) and glycosaminoglycans (GAGs) via RT-qPCR, ELISA, DMMB assay and HPLC. Mechanical loading significantly increased TNFα and PG-E2 secretion by N-SF and OA-SF, whereas in OA-SF IL-6 secretion was reduced. COL1 and FN1 secretion were downregulated in N-SF during loading. OA-SF secreted less COL1 compared to N-SF under control conditions. In contrast, OA-SF in general expressed more FN1. GAG synthesis was upregulated in N-SF, but not in OA-SF during loading with OA-SF displaying a higher charge density than N-SF. Mechanical loading enhanced proinflammatory factor expression and GAG synthesis and decreased secretion of ECM components in N-SFs, indicating a contributing role of SF to OA development.

## 1. Introduction

Osteoarthritis (OA) is the most common chronic degenerative joint disease [1] and is characterised by subchondral bone remodelling, progressive cartilage degradation, chronic pain and synovitis [2]. The pathological phenotype arises from inflammatory processes and reorganizations in the cartilage, the subchondral bone, the joint capsule and the synovial membrane [3]. Cartilage degradation and bone resorption ultimately lead to loss of joint function. The pathomechanism of OA is based on the primary destruction of extracellular matrix structures by mechanical failure and overload [4]. Articular cartilage is able to adapt to mechanical loading [5]. Moderate mechanical loading is reported to reduce expression of proteolytic enzymes and to prevent progression of OA [5]. Animal experiments showed that normal mechanical joint loading during moderate physical activity prevented cartilage degradation compared to unexercised rats [6]. Furthermore, in vitro experiments using mesenchymal stem cells revealed increased matrix deposition in response to mechanical loading [7]. Mechanical overloading, however, causes damage of the collagen network and promotes destruction of the articular cartilage, therefore triggering OA progression [5]. If the load capacity of the tissue is exceeded, a shift in the physiological balance of anabolic and catabolic processes occurs [4]. A restructuring of the cartilage matrix is already evident in early stages of the disease. The necessary matrix components like proteoglycans are proteolytically degraded, and the process of cartilage destruction begins. Chondrocytes activate catabolic enzymes [3], and the articular tissue reacts with an inflammatory reaction of the synovial membrane (synovitis). Thus, OA affects not only cartilage and subchondral bone, but also the integrity of the entire joint tissue, including the synovium. The synovium is responsible for the maintenance of synovial fluid volume and composition, mainly producing hyaluronic acid and lubricin, which helps to maintain and protect the integrity of articular cartilage [8,9]. It is composed of immune cells and synovial fibroblasts (SF). Histologically, the synovial membrane of OA joints exhibits hyperplasia of the lining cell layer and is accompanied by an infiltration of immune cells like monocytes and lymphocytes [10]. SF represent the main cell population in healthy synovium. They play a major mediating role in the development of OA, as they have immunological functions and elicit an immunological response [11]. SF have been shown to express immune receptors like toll-like receptors (TCR) [12,13]. In addition, SF were reported to produce proinflammatory cytokines and RANK-L (receptor activator of NF-κB ligand), which promotes osteoclastogenesis and therefore bone resorption [14]. There are several risk factors for developing an OA, such as aging, obesity and mechanical overload [15], as excessive mechanical loading on normal articular cartilage is generally supposed to initiate the disruption of cartilage matrix homeostasis, therefore resulting in OA [4]. Currently, however, there is limited information whether and how mechanical joint overload and consequent compressive deformation of SF also directly affect their expression profile by mechanotransductive signalling, which could possibly constitute an important additional pathomechanism of osteoarthritis. The aims of this study were thus to investigate the impact of static compressive force application on SF derived from non-OA patients (N-SF) and OA-patients (OA-SF) and to compare the effects between both groups.

## 2. Results

### 2.1. Impact of Mechanical Loading on the Expression of Proinflammatory Cytokines 

First we focused on the expression of genes and proteins involved in proinflammatory processes. In N-SF, gene and protein expression of tumour necrosis factor α (TNFα) was significantly upregulated after pressure application, whereas in OA-SF, gene expression of TNFα was not significantly altered (Figure 1a). However, an enhanced TNFα protein secretion in OA-SF was still detectable after compressive force treatment, although this protein secretion was significantly lower than in N-SF (Figure 1a). Gene expression of the proinflammatory gene cyclooxygenase-2 (COX-2) was evaluated in N-SF and in OA-SF (Figure 1b) after mechanical loading. COX-2 mRNA was significantly reduced in OA-SF under both tested conditions (Figure 1b). As COX-2 is responsible for prostaglandin-E2 (PG-E2) expression, we investigated protein levels of PG-E2 in the supernatant of stimulated cells. We found a significant induction of PG-E2 secretion in N-SF and OA-SF after compressive force application (Figure 1b). There was a slight but significant reduction of PG-E2 in OA-SF compared to N-SF under control conditions (Figure 1b). Next, we focused on the gene and protein expression of interleukin-6 (IL-6). Again, we found a significant induction of IL-6 gene expression and protein secretion in N-SF (Figure 1c) with an analogous but reduced response by OA-SF.

### 2.2. Impact of Mechanical Loading on Extracellular Matrix Composition

We investigated genes that are involved in extracellular matrix formation and remodelling. Collagen I (COL1) gene expression was significantly enhanced after mechanical loading in N-SF, but not in OA-SF (Figure 2a). Surprisingly, we found reduced COL1 secretion into the supernatant of N-SF after compressive force application. The supernatant of OA-SF already contained less COL1 under control conditions compared to N-SF (Figure 2a). Gene expression of fibronectin (FN1) was significantly upregulated after pressure treatment in N-SF and in OA-SF (Figure 2b). Like COL1, FN1 protein secretion decreased after mechanical loading in N-SF and OA-SF. OA-SF, however, secreted significantly more FN1 into the supernatant under control and pressure conditions compared to N-SF (Figure 2b). 

### 2.3. Impact of Mechanical Loading on Glycosaminoglycan (GAG) Synthesis and Sulfatation

In addition to collagen, GAGs play an important role in the composition of the extracellular matrix. We found enhanced gene expression of hyaluronan synthase 1 (HAS1) in N-SF, but not in OA-SF after mechanical loading (Figure 3a). This was accompanied by enhanced chondroitin sulphate (CS) content in N-SF after compressive force application (Figure 3b). OA-SF failed to react with enhanced CS content to mechanical loading (Figure 3b). We detected increased charge density by HPLC in N-SF after mechanical loading (Figure 3c). In OA-SF, we found a significantly increased charge density under control conditions (Figure 3c).

## 3. Discussion

In this study, we investigated the impact of mechanical loading on the expression profile of SF derived from non-OA (N-SF) and from OA patients (OA-SF) and possible differences between these conditions corresponding to pre-osteoarthritis (non-OA) and chronic osteoarthritic (OA) conditions.

In line with other studies, we demonstrated that mechanical loading on SF upregulated proinflammatory cytokines like TNFα, PG-E2 and IL-6 [16]. TNFα is considered to be a key proinflammatory cytokine during the pathological processes in the development of OA [3]. TNFα is synthesised and secreted by chondrocytes, osteoblasts, mononuclear cells and synovial fibroblasts during the inflammatory response [17,18,19]. TNFα is responsible for blocking the synthesis of proteoglycan components and collagen by chondrocytes [20,21]. In our study, mechanical loading resulted in an upregulation of TNFα protein expression in synovial fibroblasts derived from non-OA and OA patients. Of note, the reaction of SF derived from OA patients to mechanical pressure was significantly reduced. It would seem that mechanical compressive stimulation of synovial fibroblasts is more important in the initial pathogenesis of OA, but of less or no importance in the maintenance of (chronic) OA. In this phase of the disease, other cell types apart from synovial fibroblasts such as immune cells, which accumulate in the synovium during OA due to chemokines secreted by SF [22], could be of greater importance, which merits further investigation. 

TNFα can trigger an enhanced synthesis of IL-6, COX-2 and PG-E2 [23,24,25,26]. Accordingly, we could also show that the gene expression of COX-2, which is a key enzyme of PG-E2 synthesis, was increased by mechanical loading in N-SF. PG-E2 plays an important role in OA by causing inflammation, cartilage degradation and pain [27,28,29]. Mechanical loading also caused an increase in IL-6 gene expression and protein secretion in N-SF, but not in OA-SF. IL-6 is a cytokine that enhances inflammatory responses [30]. IL-6 expression causes a decrease in collagen production and an increase of matrix metalloproteinase expression [31,32,33,34]. SF derived from OA patients failed to react to the mechanical loading with enhanced IL-6 secretion, again indicating a disturbed reaction profile and hinting at a reduced importance of synovial fibroblasts in the maintenance of proinflammatory processes during osteoarthritis, in contrast to its onset.

Alterations of extracellular matrix (ECM) composition to an atypical configuration are common in OA joints [35,36]. ECM breakdown products are suggested to promote inflammation and thereby cartilage loss [37]. In our study, mechanical loading also showed an impact on extracellular matrix composition, as it impaired COL1 and FN1 protein secretion by N-SF and OA-SF. This observation is in line with reports that, in early stages of OA, collagen changes from type II to type I in the cartilage of the knee joint [38]. This compositional change affects the elasticity and mechanical stability of the ECM network [39]. Surprisingly, without mechanical loading, SF from OA-patients secreted less COL1, but more FN1 protein than N-SF. A possible explanation could be the fact that FN1 fragments were reported to induce the production of proinflammatory cytokines including TNFα [37,40], which could promote and maintain the inflammatory reaction during OA. 

Hyaluronic acid (HA) is synthesized by HA synthase 1 (HAS1) at the inner surface of the plasma membrane [41]. It serves a variety of functions, including lubrication of joints [22]. Changes in the serum concentration of HA are associated with inflammatory and degenerative arthropathies such as rheumatoid arthritis [42,43]. In a murine model of knee joint cartilage damage, a deficiency of HAS1 triggered chronic joint inflammation and was associated with widespread intra-articular fibrosis [44]. Chondroitin sulphate is a sulphated heteropolysaccharide found in the cartilage and extracellular matrix [45,46]. CS is negatively charged and reported to interact readily with extracellular matrix proteins modulating cellular activities [47]. In this study, mechanical loading significantly increased charge density, chondroitin sulphate (CS) content as well as gene expression of HAS1 in N-SF, but not in OA-SF, again indicating an impaired responsiveness to mechanical loading in line with the previous observations. CS are the most widely used drugs for the treatment of OA [48] and are reported to slow down OA development [49]. The enhanced CS content measured in N-SF due to mechanical loading may therefore provide a protection mechanism for the joint.

Judging from these observations, it would seem that the combination of inflammation-induced downregulation of ECM synthesis and proinflammatory effects of ECM-breakdown-components like FN1 promotes OA onset and progression induced by mechanical loading.

## 4. Materials and Methods 

### 4.1. In Vitro Cell Culture Experiment Setup

We obtained synovial fibroblasts either from non-OA patients (N-SF, *n* = 2; male and female; age: 39 and 47) or from OA patients (OA-SF, *n* = 3, two females, one male; age: 67–76). N-SF were ordered from BIOIVT (PCD-90-0645, West Sussex, UK). OA-SF were derived and cultivated from excess tissue arising during knee surgery of patients with osteoarthritis in the Department of Orthopedics of the University of Regensburg. Briefly, we released synovia from surrounding fat and shred the tissue using a scalpel in a dish with 1 mL dispase I (D4818, Sigma-Aldrich, Munich, Germany). We incubated tissue slices for 2 h in dispase I at 37 °C. Supernatant was put onto a 70 µm cell strainer, dispase I removed by centrifugation and cells were grown in DMEM high glucose (D5796, Sigma-Aldrich) supplemented with 10% FCS (P30-3306, PAN-Biotech, Aidenbach, Germany), 1% *L*-glutamine (SH30034.01, GE Healthcare Europe, Munich, Germany), 100 µM ascorbic acid (A8960, Sigma-Aldrich) and 1% antibiotics/antimycotics (A5955, Sigma-Aldrich). All synovial fibroblasts were characterized according to their spindle-shaped morphology and the expression of fibroblast specific genes. For our experiments, we seeded 60,000 synovial fibroblasts per well onto standard 6-well cell culture plates and preincubated the cells for 24h at standard cell culture conditions (37 °C, 5% CO_2_, 100% H_2_O) in DMEM high glucose, supplemented as already indicated above. To minimize the risk of contamination, particularly with immune cells, we used fibroblasts not earlier than the 3rd passage [50]. Fibroblasts were then further incubated for another 48 h and either left untreated (controls) or compressed with a pressure level of 2 g/cm^2^ by applying a sterile glass disc according to an established and published model [51,52,53,54] (Figure 4) to simulate mechanical loading. After 72 h of incubation, we harvested the cells and the cell culture supernatant for further analyses.

### 4.2. RNA Isolation and RT-qPCR

We performed RNA isolation and quality assessment as well as RT-qPCR, as described before according to MIQE (minimum information for publication of quantitative real-time PCR experiments) guidelines [51]. Briefly, we extracted total RNA from synovial fibroblasts using 1 mL peqGOLD TriFast™ (PEQLAB) per well, and further processing was conducted according to the manufacturer’s instructions. We eluted the obtained RNA pellet in 20 µL nuclease-free water. As shown before, this extraction protocol guarantees sufficient RNA integrity (RIN, 28S/18S ratio) as well as absence of genomic DNA contamination [51]. We transcribed a standardized amount of RNA per sample by using 0.5 µL of an oligo-dT18 primer (SO131, Thermo Fisher Scientific, Waltham, MA, USA), 0.5 µL of a random hexamer primer (SO142, Thermo Fisher Scientific), 0.5 µL dNTP mix (L785.2, Carl Roth, Karlsruhe, Germany), 0.5 µL of an RNase inhibitor (EO0381, Thermo Fisher Scientific), 0.5 µL reverse transcriptase (M1705, Promega, Madison, WI, USA) and 2 µL 5× M-MLV-buffer (M1705, Promega). Nuclease-free H_2_O was added to a total volume of 10 µL for cDNA synthesis. We incubated this reaction mixture for 60 min at 37 °C. After heat inactivation of reverse transcriptase (95 °C, 2 min), we diluted cDNA 1:10 with nuclease-free H_2_O and stored it at −20 °C until use. cDNA synthesis was performed for all samples at the same time to minimize experimental variation. RT-qPCR amplification was performed with a Mastercycler^®^ ep realplex-S thermocycler (Eppendorf, Hamburg, Germany), as described before [51]. Each reaction mix contained 7.5 µL SYBR^®^ Green JumpStart™ Taq ReadyMix™ (S4438; Sigma–Aldrich), 0.75 µL of the respective primer pair (3.75 pmol/primer) and 1.5 µL of the respective cDNA dilution. Nuclease-free H_2_O was added to a total amount of 15 µL. We prepared all components except the cDNA solution as a master-mix to avoid technical errors during manual pipetting. cDNA amplification was performed in 45 cycles (initial heat activation 95 °C/5 min, per cycle 95 °C/10 s denaturation, 60 °C/8 s annealing, 72 °C/8 s extension). We used a set of two reference genes (EEF1A1 and RPLP0) for normalization of target genes (relative gene expression), which have been tested and shown to be stably expressed in synovial fibroblasts under the investigated conditions (data not shown). Relative gene expression was calculated as 2-Δ*C*q [55]. We constructed all intron-flanking, gene-specific primers (Table 1) according to MIQE quality guidelines and criteria as described before [51]. 

### 4.3. ELISA Assays

For quantification of tumour necrosis factor α (TNFα) and prostaglandin E2 (PGE2), interleukin-6 (IL-6), collagen-1 (COL-1) and fibronectin (FN-1) protein secretion into the synovial fibroblast supernatant, we used commercially available ELISA kits according to the manufacturers’ instructions (TNFα: EK0525, Boster Biological Technology; PGE2: 514010; Cayman chemical; IL-6: EK0410 Boster Biological Technology; COL-1: ab210966, Abcam; FN-1: EK0349, Boster Biological Technology). Protein expression per well was related to the respective number of synovial fibroblasts, counted with a Beckman Coulter Counter (Z2 cell counter).

### 4.4. Isolation of Glycosaminoglycans (GAGs)

Approximately 1.5 mL cell culture supernatant was freeze-dried at the end of incubation. After reconstitution of the pellet in 300 µL H_2_O, we added 900 µL ethanol and incubated at −20 °C overnight. Samples were then centrifuged for 5 min at 5000 rpm, and the resulting pellet was solved in 300 µL 0.1 M NH_4_Ac. Samples were incubated with 20 U/mL proteinase K (Sigma-Aldrich) for 2 h at 55 °C. After heat inactivation (5 min, 100 °C), we added 900 µL ethanol and incubated overnight at −20 °C. After centrifugation for 5 min at 5000 rpm, the pellet was reconstituted in 50 µL H_2_O and used for further analyses (DMMB and HPLC).

### 4.5. Determination of Chondroitin Sulphate Content via DMMB Assay

DMMB (dimethylenmethylblue, Sigma-Aldrich) specifically stains chondroitin sulphate. Ten microlitres of the respective eluted GAG sample or of a standard dilution series (1:10) of known concentrations of chondroitin sulphate A (Sigma-Aldrich), used for calibration, were mixed with 190 μL DMMB staining solution (10 mL 2 M NaCl, 1.5 g glycine solved in 490 mL H_2_O, pH 3.0 adjusted with HCl; 8 mg DMMB solved in 2.5 mL ethanol and added) and measured in an ELISA-reader (Multiscan Go, Thermo Fisher Scientific) at 520 nm. All measurements were performed in triplets. 

### 4.6. HPLC Analysis

We treated 10 µL of the GAG samples with 5 mU of chondroitin ABC lyase (Sigma-Aldrich) and incubated them in 50 µL of 100 mM Tris / 150 mM sodium acetate buffer (pH 8.0) at 37 °C for 16 h. The reaction was blocked by boiling the solution for 1 min. We analysed the unsaturated disaccharides generated from hyaluronic acid (HA) and chondroitin sulphate (CS) after enzymatic treatment of the purified GAGs by strong anion exchange HPLC separation at 232 nm (column: Sphere-Image 80-5 SAX, Knauer, Berlin, Germany; equipment: Shimazdu, Duisburg, Germany, consisting of degasifier DGU-20A3, two pumps LC-20AT, autosampler SIL-20A, controller CBM-20A, detector SPD-20A; software: LCSolution). Isocratic separation was performed for 15 min with 10 mM NaH_2_PO_4_, pH 4.0 and afterwards linear gradient separation for another 20 min with 10 mM NaH_2_PO_4_, pH 4.0 to 33% 750 mM NaH_2_PO_4_, pH 4.0. Flow rate was 1.2 mL/min. Charge density was calculated by dividing the area under the curve (AUC) for CS by the AUC for (HA+CS).

### 4.7. Statistical Analysis

To obtain normalized data values we divided all absolute data by the respective arithmetic mean of the pressure-untreated N-SF controls prior to statistical analysis. All data were tested for normal distribution (Shapiro-Wilk test) and homogeneity of variance (Levene’s test) using the software SPSS^®^ Statistics 24 (IBM^®^, Armonk, NY, USA). Descriptive statistics are given as mean ± standard deviation. The experimental groups were independently compared by one-way ANOVAs, which were validated by applying Welch’s test, since homogeneity of variance not always present. For the same reason, we used Games–Howell post hoc tests for heterogeneous variances for pairwise comparisons. All differences were considered statistically significant at *p* ≤ 0.05.

## 5. Conclusions 

In our study, SF react to mechanical loading with enhanced expression and secretion of proinflammatory cytokines and changes of ECM composition. These changes might contribute to OA pathogenesis and progression. SF derived from OA patients seem to be less responsive to mechanical loading, indicating a more important role of SF in OA onset than in OA maintenance, which might be predominantly mediated by cells of the immune system. Moreover, it should be noted that OA pathogenesis and progression are complex and multicellular processes; thus, the results derived from SF need to be corroborated for other OA-relevant cell types such as immune cells in future experiments. Furthermore, different mechanical loading protocols might elucidate the role of SF in catabolic and anabolic events during OA.

## Figures and Tables

**Figure 1 ijms-20-00585-f001:**
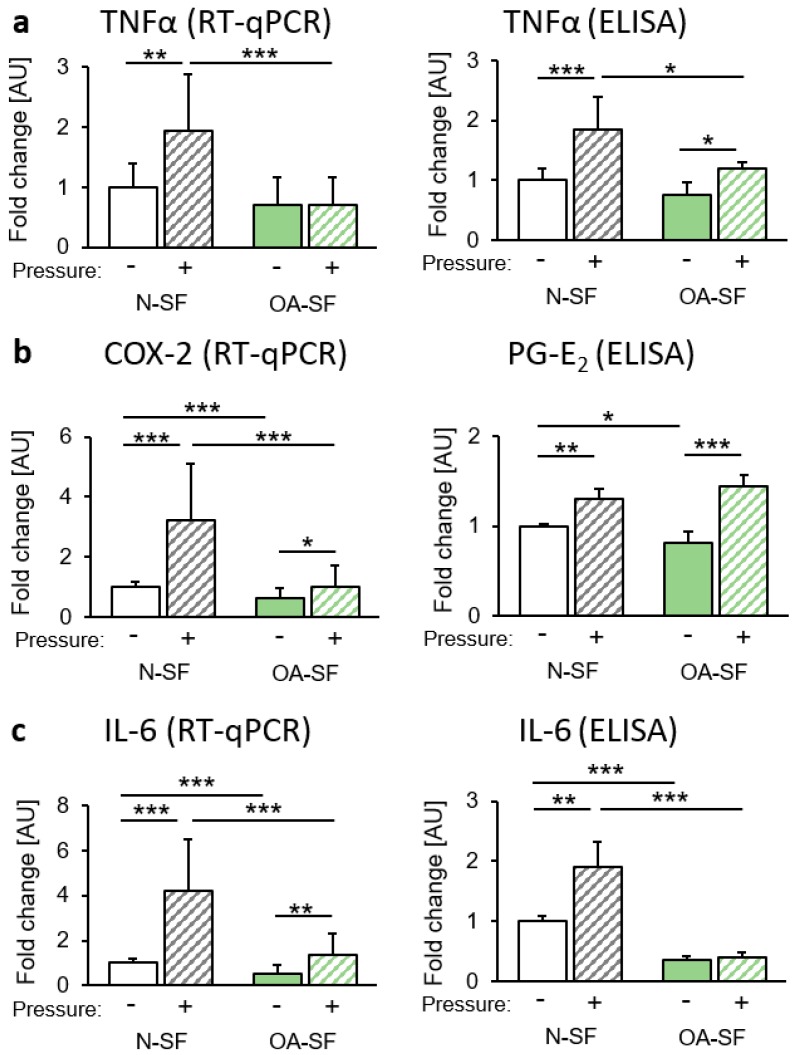
Impact of static compressive force application on the expression of proinflammatory genes. Gene expression and protein secretion of (**a**) TNF-α, (**b**) COX-2/PG-E2 and (**c**) IL-6 of N-SF and OA-SF after 48 h with or without static compressive force application. AU: arbitrary units; RT-qPCR: *n* = 9; ELISA: *n* = 6. * *p* ≤ 0.05, ** *p* ≤ 0.01, *** *p* ≤0.001. Statistics: Welch-corrected ANOVA with Games-Howell post-hoc-tests.

**Figure 2 ijms-20-00585-f002:**
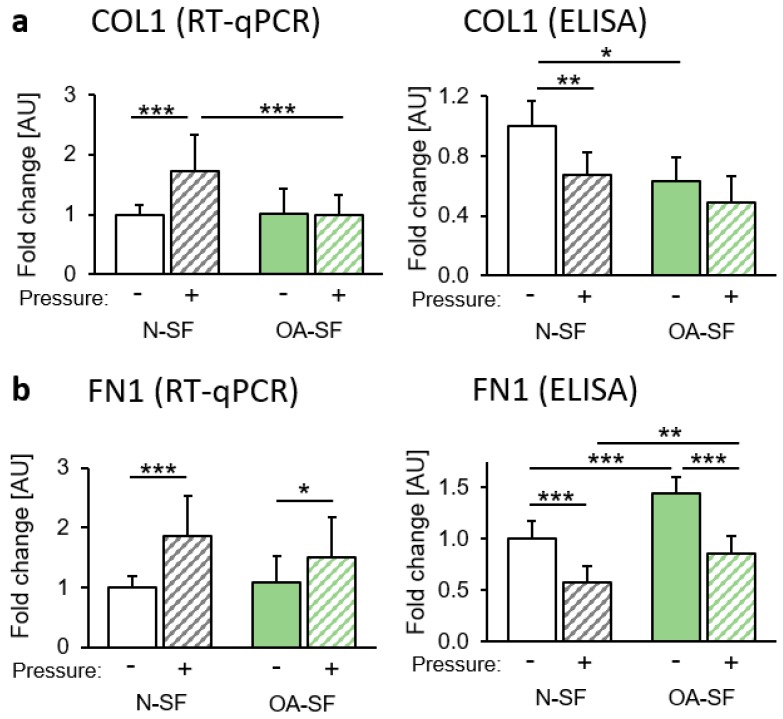
Impact of static compressive force application on the extracellular matrix. Gene expression and protein secretion of (**a**) COL1 and (**b**) FN1 of N-SF and OA-SF after 48 h with or without static compressive force application. AU: arbitrary units; RT-qPCR: *n* = 9; ELISA: *n* = 6. * *p* ≤ 0.05, ** *p* ≤ 0.01, *** *p* ≤ 0.001. Statistics: Welch-corrected ANOVA with Games-Howell post-hoc-tests.

**Figure 3 ijms-20-00585-f003:**
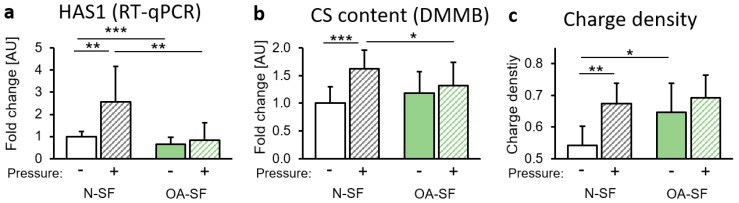
Impact of static compressive force application on glycosaminoglycans (GAG) synthesis. (**a**) Gene expression of hyaluronan synthase 1 (HAS1) by N-SF and OA-SF after 48 h with or without static compressive force application. (**b**) Chondroitin sulphate (CS)-content (DMMB assay) and (**c**) charge density (HPLC) of GAGs of N-SF and OA-SF after 48 h with or without static compressive force application. AU: arbitrary units; RT-qPCR: *n* = 9; DMMB: *n* = 9. * *p* ≤ 0.05, ** *p* ≤ 0.01, *** *p* ≤ 0.001. Statistics: Welch-corrected ANOVA with Games-Howell post-hoc-tests.

**Figure 4 ijms-20-00585-f004:**
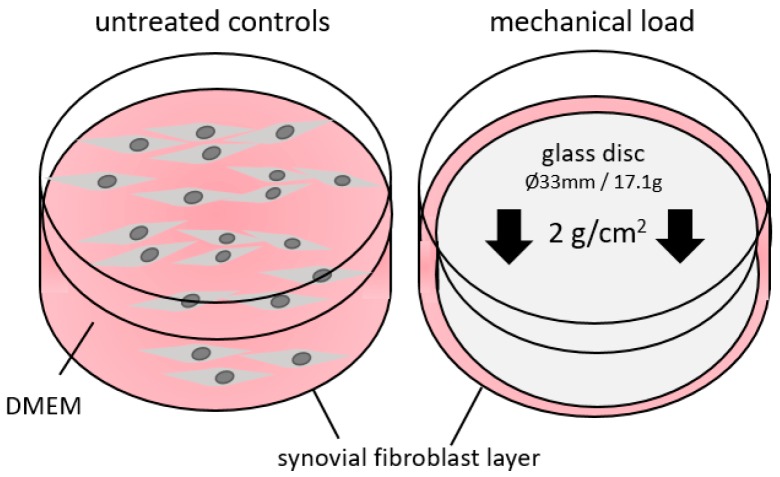
Model of static compressive force application. Synovial fibroblasts either derived from non-OA patients (N-SF) or from OA patients were treated with static compressive force for 48 h under normal cell culture conditions. DMEM—Dulbecco‘s modified eagle medium.

**Table 1 ijms-20-00585-t001:** Reference and target genes used for RT-qPCR.

Gene Symbol	Gene Name (Homo Sapiens)	Accession Number	5′-Forward Primer-3′	5′-Reverse Primer-3′
EEF1A1	eukaryotic translation elongation factor 1 α1	NM_001402.5	CCTGCCTCTCCAGGATGTCTAC	GGAGCAAAGGTGACCACCATAC
RPLP0	ribosomal protein, large, P0	NM_001002.3	GAAACTCTGCATTCTCGCTTCC	GACTCGTTTGTACCCGTTGATG
B4GALNT	β-1,4-*N*-acetyl-galactosaminyl transferase 4	NM_178537	GAAGATCCGTAAGCAGATGAAGC	ACGGCTCTCACTGGAGTCC
COL1	collagen, type I, α2	NM_000089.3	AGAAACACGTCTGGCTAGGAG	GCATGAAGGCAAGTTGGGTAG
FN1	fibronectin 1	NM_212482.1	GCCAGTCCTACAACCAGTATTCTC	GCTTGTTCCTCTGGATTGGAAAG
HAS1	hyaluronan synthase 1	NM_001523.3	GAGCCTCTTCGCGTACCTG	CCTCCTGGTAGGCGGAGAT
IL6	interleukin 6	NM_000600.3	TGGCAGAAAACAACCTGAACC	CCTCAAACTCCAAAAGACCAGTG
COX2	prostaglandin-endoperoxide synthase 2	NM_000963.3	GAGCAGGCAGATGAAATACCAGTC	TGTCACCATAGAGTGCTTCCAAC
TNFα	Tumor necrose factor α	NM_000594.3	GAGGCCAAGCCCTGGTATG	CGGGCCGATTGATCTCAGC

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
