# Peer review of "Impact of Mechanical Load on the Expression Profile of Synovial Fibroblasts from Patients with and without Osteoarthritis"

_ijms, 2019, doi:10.3390/ijms20030585_

Reviewer 1 Report

This is a nicely presented manuscript with new and original data. Of course, actual relevance to the in vivo OA disease scenario is inferred, but this is fine because the study employs an in vitro model. The data is nicely presented and the statistical analyses have been well done.

Reviewer 2 Report

Manuscript titled “Impact of mechanical load on the expression profile of synovial fibroblasts from patients with and without osteoarthritis” deals an important issue of medical cartilage biology. In this paper, the authors investigated the effects of mechanical loading on SF derived from non-OA (N-SF) and OA patients (OA-SF). 

The paper is good and well written. Before to accept it for publication need some mandatory revisions.

Abstract: 

please delete the following inappropriate sentence: Mechanical loading is a major risk factor in OA pathogenesis or fix it better adding eg. excessive mechanical loading..

Introduction: 

please fix the reference number in the text in order to the citation.

Please add the benefical effect of Mechanical Loading and moderate physical activity on Articular Cartilage, please discuss and quote the following papers:

·     Asymmetrical seeding of MSCs into fibrin-poly(ester-urethane) scaffolds and its effect on mechanically induced chondrogenesis. J Tissue Eng Regen Med. 2017 Oct;11(10):2912-2921.

·     The Effect of Mechanical Loading on Articular Cartilage. J. Funct. Morphol. Kinesiol. 2016, 1, 154-161.

·     Physical activity ameliorates cartilage degeneration in a rat model of aging: a study on lubricin expression. Scand J Med Sci Sports. 2015 Apr;25(2):e222-30.

Pag. 2, lines 52-54, please add after hyaluronic acid “and lubricin”. And quote the following citation: Biosynthesis of collagen I, II, RUNX2 and lubricin at different time points of chondrogenic differentiation in a 3D in vitro model of human mesenchymal stem cells derived from adipose tissue. Acta Histochem. 2014 Oct;116(8):1407-17.

In the conclusion please specify better the clinical relevance of your work, limitation, and some important suggestions for the scientific community.

Author Response

Manuscript titled “Impact of mechanical load on the expression profile of synovial fibroblasts from patients with and without osteoarthritis” deals an important issue of medical cartilage biology. In this paper, the authors investigated the effects of mechanical loading on SF derived from non-OA (N-SF) and OA patients (OA-SF). 

The paper is good and well written. Before to accept it for publication need some mandatory revisions.

Abstract:  please delete the following inappropriate sentence: Mechanical loading is a major risk factor in OA pathogenesis or fix it better adding eg. excessive mechanical loading..

We change the sentence to: Excessive mechanical loading might contribute to OA pathogenesis

Introduction: please fix the reference number in the text in order to the citation.

We fixed the reference numbers in the text in order to the citation order.

Please add the benefical effect of Mechanical Loading and moderate physical activity on Articular Cartilage, please discuss and quote the following papers:

Asymmetrical seeding of MSCs into fibrin-poly(ester-urethane) scaffolds and its effect on mechanically induced chondrogenesis. J Tissue Eng Regen Med. 2017 Oct;11(10):2912-2921.

The Effect of Mechanical Loading on Articular Cartilage. J. Funct. Morphol. Kinesiol. 2016, 1, 154-161.

Physical activity ameliorates cartilage degeneration in a rat model of aging: a study on lubricin expression. Scand J Med Sci Sports. 2015 Apr;25(2):e222-30.

We added the suggested articles to the introduction and discussed them according your suggestions.

Articular cartilage is able to adapt to mechanical loading [5]. Moderate mechanical loading is reported to reduce expression of proteolytic enzymes and to prevent progression of OA [5]. Animal experiments showed that normal mechnical joint loading during moderate physical activity prevented cartilage degradation compared to unexpercised rats [6]. Furthermore in vitro experiments using mesenchymal stem cells revealed increased matrix deposition in response to mechnical loading [7]. Mechnical overloading however causes damage of the collagen network and promotes destruction of the articular cartilage therefore triggering OA progression [5].

Pag. 2, lines 52-54, please add after hyaluronic acid “and lubricin”. And quote the following citation: Biosynthesis of collagen I, II, RUNX2 and lubricin at different time points of chondrogenic differentiation in a 3D in vitro model of human mesenchymal stem cells derived from adipose tissue. Acta Histochem. 2014 Oct;116(8):1407-17.

We added lubricin and cited the suggested article.

In the conclusion please specify better the clinical relevance of your work, limitation, and some important suggestions for the scientific community.

We revised the conclusion according your suggestions:

In our study SF react to mechanical loading with enhanced expression and secretion of proinflammatory cytokines and changes of ECM composition. These changes might contribute to OA pathogenesis and progression. SF derived from OA patients seem to be less responsive to mechanical loading, indicating a more important role of SF in OA onset than in OA maintenance, which might be predominantly mediated by cells of the immune system. Also it should be noted that OA pathogenesis and progression are complex and multicellular processes, thus the results derived from SF need to be corroborated for other OA-relevant cell types such as immune cells in future experiments. Furthermore different  mechanical loading protocols might elucidate the role of SF in catabolic and anabolic events during OA in the future.

 Reviewer 3 Report

The manuscript by Schroder et al. titled "Impact of mechanical load on the expression profile of synovial fibroblasts from patients with and without osteoarthritis" investigated on the effects of mechanical load on synovial fibroblast expression profile.

The authors showed that mechanical load increased TNF-a, and PG-E2 production by both non-osteoarthritis (N-SF) and osteoarthritis derived synovial fibroblast (OA-SF). Moreover, mechanical load increased the expression of pro-inflammatory factors in N-SF.

The manuscript is well written and the experiments were well planned. However, some points should be addressed or better clarified. It is not completely clear how the authors obtain synovial fibroblasts for the in vitro experiments, this an important point to know whether immune cells "contaminate" synovial fibroblast culture and whether they influence expression profile. Related to this issue, which is the impact of mechanical load on cytokine/chemokine profile immune cells in vitro? This point should be addressed.

In addition, it is interesting to know whether supernatants collect from N-SF undergo to mechanical load were able to modify expression profile of fibroblasts which do not undergo the mechanic stress.

Author Response

The manuscript by Schroder et al. titled "Impact of mechanical load on the expression profile of synovial fibroblasts from patients with and without osteoarthritis" investigated on the effects of mechanical load on synovial fibroblast expression profile.

The authors showed that mechanical load increased TNF-a, and PG-E2 production by both non-osteoarthritis (N-SF) and osteoarthritis derived synovial fibroblast (OA-SF). Moreover, mechanical load increased the expression of pro-inflammatory factors in N-SF.

The manuscript is well written and the experiments were well planned. However, some points should be addressed or better clarified.

It is not completely clear how the authors obtain synovial fibroblasts for the in vitro experiments, this an important point to know whether immune cells "contaminate" synovial fibroblast culture and whether they influence expression profile.

We revised Material & Methods section and added a paragraph for isolation of synovial fibroblasts from OA-patients.

Briefly, we released synovia from surrounding fat and shred the tissue using a scalpel in dish with 1 ml dispase I (D4818, Sigma-Aldrich). We incubated tissue slices for 2 h in dispase I at 37°C. Supernatant was put onto a 70µm cell strainer, dispase I removed by centrifugation and cells were grown in DMEM high glucose (D5796, Sigma-Aldrich, Munich, Germany), supplemented with 10% FCS (P30-3306, PAN-Biotech, Aidenbach, Germany), 1% L-glutamine (SH30034.01, GE Healthcare Europe, Munich, Germany), 100 µM ascorbic acid (A8960, Sigma-Aldrich) and 1% antibiotics/antimycotics (A5955, Sigma-Aldrich). All synovial fibroblasts were characterized according to their spindle-shaped morphology and the expression of fibroblast specific genes.   For our experiments we seeded 60.000 synovial fibroblasts per well onto standard 6-well cell culture plates and preincubated the cells for 24h at standard cell culture conditions (37°C, 5% CO2, 100% H2O) in DMEM high glucose, supplemented as already indicated above. To minimize the risk of contamination particularly with immune cells, we used fibroblast not earlier than the 3rd passage [47].

Related to this issue, which is the impact of mechanical load on cytokine/chemokine profile immune cells in vitro? This point should be addressed.

We are actually working on performing experiments with mechanical loading on macrophages. But as already mentioned, a contamination of fibroblasts with immune cells seems negligible due to the characterization of fibroblasts by their morphology and specific expression pattern, particularly from the 3rd passage onwards as has been shown before by Zhao J, Ouyang Q, Hu Z et al. (2016) in “A protocol for the culture and isolation of murine synovial fibroblasts.” Biomed Rep 5(2): 171–175. doi: 10.3892/br.2016.708. We added this aspect to the conclusions section: “Also it should be noted that OA pathogenesis and progression are complex and multicellular processes, thus the results derived from SF need to be corroborated for other OA-relevant cell types such as immune cells in future experiments.”

In addition, it is interesting to know whether supernatants collect from N-SF undergo to mechanical load were able to modify expression profile of fibroblasts which do not undergo the mechanic stress.

This is an interesting point worthy of further investigation! Unfortunately, we didn´t perform this experiment for this study and cannot address this question within the five days alloted by the journal for the minor revision of the manuscript and the reponse to the reviewers’ comments. But we will have this in mind for further studies, as we progress in our research on OA. As both proinflammatory cytokines and the extracellular matrix are affected on the protein level (cell culture supernatants) it seems likely that TNFalpha for instance will change the expression pattern of synovial fibroblasts, which will be investigated by our group in a future study.